

# Two new free-living marine species of *Desmodorella* (Nematoda: Desmodoridae) from the continental shelf of northeastern Brazil, with an emended generic diagnosis and a dichotomous key to the species

Alex Manoel[1,2], Patricia F. Neres[1] and André M. Esteves[1,†]

[1] Zoologia, Universidade Federal de Pernambuco, Recife, PE, Brazil
[2] Biologia Animal Graduate Course, Universidade Federal de Pernambuco, Recife, PE, Brazil
[†] Deceased.

## ABSTRACT

Two new species of *Desmodorella* are described from sediment samples collected on the continental shelf of northeastern Brazil. Although the occurrence of the genus has been previously reported in this region, the present study provides the first formal descriptions of *Desmodorella* species based on material collected from the Brazilian coast. *Desmodorella cornuta* sp. nov. is characterized by a protuberant horn-shaped cuticular projection positioned dorsally in the pharyngeal region, a unique characteristic among species of the genus. *Desmodorella parabalteata* sp. nov. is distinguished from other species by the presence of a cephalic capsule and cuticle ornamented with vacuoles, multispiral amphidial fovea, longitudinal rows of ridges that are often indistinct under light microscopy, two pairs of lateral rows of spines that are more prominent than the remaining rows, and thin, nearly straight spicules with a slightly swollen proximal end and lacking a capitulum. An emended diagnosis of the genus and a dichotomous key to species are provided.

## INTRODUCTION

The Family Desmodoridae *Filipjev, 1922* currently includes six subfamilies, 50 genera, and approximately 430 species (*Nemys eds., 2025*). The available literature on the genus *Desmodorella Cobb, 1933* documents several taxonomic revisions that have altered its status over time (*Cobb, 1933*; *Gerlach, 1950*; *Wieser, 1954*; *Gerlach, 1963*; *Lorenzen, 1976*; *Verschelde, Gourbault & Vincx, 1998*).

After its establishment by *Cobb (1933)*, *Gerlach (1950)* suggested that *Desmodorella* might represent a subgenus of *Desmodora De Man, 1889*. However, he merely noted this possibility without formally reclassifying the taxon. Subsequently, *Wieser (1954)*, based on *Gerlach (1950)*, reduced *Desmodorella* to a subgenus of *Desmodora*. Similarly, *Gerlach*

Corresponding author
Alex Manoel, alexblg08@gmail.com

*(1963)* in his review of the Desmodoridae, considered *Desmodorella* as a subgenus of *Desmodora*. He argued that several genera closely related to *Desmodora*, described earlier by *Cobb (1920)* and *Cobb (1933)*, should in fact be treated as subgenera of *Desmodora*. In the same study, *Gerlach (1963)* provided a key to the subgenera, using the morphology of the amphidial fovea as the primary diagnostic character.

*Lorenzen (1976)* disagreed with *Gerlach (1963)*, arguing that amphidial fovea morphology alone was insufficient for differentiating the subgenera *Desmodora* and *Desmodorella*, and thus synonymized *Desmodorella* with *Desmodora*. In contrast, *Verschelde, Gourbault & Vincx (1998)* revised the genus *Desmodora* and, while concurring that amphidial fovea morphology was not a reliable distinguishing character, disagreed with the synonymization proposed by Lorenzen. *Verschelde, Gourbault & Vincx (1998)* reconsidered *Desmodorella* as a valid genus within the Desmodoridae and argued that *Desmodorella* can be easily distinguished from *Desmodora* by the presence of longitudinal rows of ridges or spines along the body—a feature consistently present in *Desmodorella* but absent in *Desmodora*. Additionally, species of *Desmodorella* were noted to have spicules either lacking a capitulum or bearing only a minute one, and without a velum.

Marine representatives of *Desmodorella* have been recorded from the Pacific (*Verschelde, Gourbault & Vincx, 1998*), Atlantic (*Gerlach, 1950*; *Riera, Núñez & Brito, 2012*), Indian (*Annapurna et al., 2012*), and Antarctic Oceans (*Ingels et al., 2006*), with habitats ranging from the intertidal zone (*Riera, Núñez & Brito, 2012*) to deep-sea environments (*Verschelde, Gourbault & Vincx, 1998*; *Fadeeva, Mordukhovich & Zograf, 2016*). Occurrences of this genus have also been recorded in freshwater bodies (*Gagarin & Nguyen, 2003*; *Decraemer & Smol, 2006*). Along the Brazilian coast, the occurrence of this genus was recorded in dissertations/theses for deep-sea regions in the Campos Basin, Southeast Brazil (*Silva, 2012*; *Moura, 2013*) and for the Continental Shelf of the Potiguar Basin, Northeast Brazil (*Larrazábal-Filho, 2020*).

The present study reports on specimens of *Desmodorella* from the continental shelf of northeastern Brazil, describes two new species, and updates the generic diagnosis by incorporating new morphological characters. Here we also propose a dichotomous key based on male characteristics to facilitate the identification of *Desmodorella* species. Additionally, we highlight the key diagnostic traits that should be jointly considered for accurate species-level differentiation within the genus.

## MATERIAL AND METHODS

**Study area and sampling (Table 1).** The sediment used in the study of these animals was obtained from two projects that conducted sampling at different stations along the coast of northeastern Brazil. Table 1 presents details of the collection stations relevant to this study. In both projects the sediment samples were taken in triplicate. For sediment collection, a box-corer or Van Veen grab was used (see Table 1), while meiofauna samples were collected with a 10 cm × 10 cm corer. The collected material was transferred into plastic containers and preserved in a 4% buffered formaldehyde solution.

**Laboratory processing.** In the laboratory, sediment samples were sieved using a 0.500 mm mesh, followed by a 0.045 mm mesh to retain meiobenthic organisms. The material

**Table 1  Collection stations, respective coordinates, depth, collection gear, and environmental characteristics. The samples were collected from the continental shelf in northeastern Brazil, South Atlantic.** Environmental characteristics: median grain size (mm), CaCO₃ (%), organic matter (%).

| Project | Station | Sampling date | Latitude (S) | Longitude (W) | Depth (m) | Gear | Median grain size (mm) | CaCO₃ (%) | Organic matter (%) |
|---------|---------|---------------|--------------|---------------|-----------|------|------------------------|-----------|---------------------|
| Evaluation of benthic and planktonic biota in the offshore portion of the Potiguar and Ceará basins | ME2B2 R1 | 06/09 | 05°02′29.6″ | 36°23′11.9″ | 8.5 | Van Veen grab | 0.46 | 33 | 4.2 |
| | ME2B2 R3 | 06/13 | 05°02′30.3″ | 36°23′12.3″ | 8.5 | | 0.44 | 54 | 6.9 |
| | ME2B3 R2 | 06/12 | 05°01′12.4″ | 36°23′27.6″ | 8.1 | | 0.61 | 39 | 4.7 |
| UFPE S.O.S. SEA | 14 | 09/12/2019 | 10°07′05.7″ | 35°50′57.96″ | 63 | Box-corer | 0.031 | 87 | 3.5 |
| | 16 | 09/12/2019 | 10°44′59.28″ | 36°25′32.88″ | 58 | | 0.003 | 81 | 6.1 |
| | 17 | 10/12/2019 | 11°00′00.54″ | 36°49′58.98″ | 54 | | 0.003 | 67 | 2.9 |
| | 23 | 11/12/2019 | 13°04′10.32″ | 38°25′46.98″ | 65 | | 0.003 | 42 | 5.9 |

retained on the 0.045 mm mesh was subsequently extracted using SICOL-40 colloidal silica solution (specific gravity 1.18) (*Somerfield, Warwick & Moens, 2005*).

Nematodes were counted and extracted under a stereomicroscope using a Dollfus plate. Each specimen was subsequently placed into a small glass container filled with a solution comprising formaldehyde (4%) with glycerin (1%) (Solution 1—*De Grisse, 1969*). The procedure for transferring each organism to glycerin was implemented, followed by clearing in accordance with the method outlined by *De Grisse (1969)*. The specimens were then permanently mounted on glass slides, following the technique described by *Cobb (1920)*. The genus was identified using the identification keys provided by *Platt & Warwick (1988)* as well as *Decraemer & Smol (2006)*. Species identification was achieved by comparing their characteristics with those detailed in the original descriptions. Illustrations were created with the assistance of an Olympus CX31 optical microscope equipped with a drawing tube. Body measurements were recorded using a mechanical map meter.

For scanning electron microscopy (SEM), specimens were taken from previously mounted glycerin-paraffin slides. These specimens underwent rehydration using distilled water, following the protocol outlined by *Abolafia (2015)*. Subsequently, the specimens were transferred to a meiofauna processing container, as described by *Abolafia (2015)*, and subjected to a gradual dehydration process through a series of graded ethanol concentrations (10% for one day, followed by 20%, 30%, 40%, 50%, 60%, 70%, 80%, 90%, 92%, 95%, and two rounds of 100% on the second day, with transitions between concentrations occurring every two hours). After dehydration, the specimens were dried using a critical point dryer. Finally, the specimens were removed from the container, placed on an aluminum stub that was covered with conductive tape, sputter-coated with gold, and examined using a TM4000 SEM at 10 kV with a backscattered electron (BSE) detector or by combining this with the secondary electron (SE) detector.

The holotype and a female paratype of each species are deposited in the Nematoda Collection at the Museum of Oceanography Prof. Petronio Alves Coelho (MOUFPE), Brazil. Additional paratypes are stored in the Meiofauna Laboratory, Department of Zoology at the Federal University of Pernambuco (NM LMZOO-UFPE).

The digital version of this article, presented in Portable Document Format (PDF), constitutes a published study in compliance with the standards established by the International Commission on Zoological Nomenclature (ICZN). As a result, the new names introduced in this digital edition are considered effectively published under the Code, relying exclusively on the electronic format. This work, along with its nomenclatural acts, has been duly registered in ZooBank, the online registration system of the ICZN.ZooBank Life Science Identifiers (LSIDs) are available and can be viewed in any standard web browser by appending the LSID to the prefix http://zoobank.org/. The LSID for this publication is: urn:lsid:zoobank.org:pub:0EC65900-F3B5-4486-B067-5721DAC18C4D. The online version of this research is archived and accessible through the following digital repositories: PeerJ, PubMed Central, and CLOCKSS.

## RESULTS

### Systematics

**Taxonomic classification, according to De Ley & Blaxter (2004)**
**Class Chromadorea** *Inglis, 1983*
**Subclass Chromadoria** *Pearse, 1942*
**Order Desmodorida** *De Coninck, 1965*
**Suborder Desmodorina** *De Coninck, 1965*
**Superfamily Desmodoroidea** *Filipjev, 1922*
**Family Desmodoridae** *Filipjev, 1922*
**Subfamily Desmodorinae** *Filipjev, 1922*
**Genus** *Desmodorella* *Cobb, 1933*

**Diagnosis.** (Emended from *Leduc & Zhao, 2016*) Annulated cuticle ornamented with longitudinal rows of ridges or hair-like spines (sometimes indistinct under light microscope). Cuticular vacuoles present or absent. Lateral alae absent. Two pairs of lateral rows with more distinct spines, among the other rows of spines ("false lateral alae") present or absent. Two to eight longitudinal rows of somatic setae. Horn-shaped cuticular projections present or absent. Head capsule truncated or rounded, either smooth or ornamented with numerous vacuoles (appearing smooth or wrinkled under SEM). Cephalic setae anterior to or at level of anterior edge of amphidial fovea. Rows of subcephalic setae present or absent (additional scattered setae may occur without forming distinct rows). Large multispiral to loop-shaped amphidial fovea located on head capsule (with the largest portion situated on the labial region in *D. spineacaudata*). Pharynx with a rounded or oval posterior bulb. Males with one anteriorly outstretched testis. Spicules slender or filiform, short to elongated, lacking a prominent capitulum and velum. Gubernaculum present, with or without lateral pieces (crurae). Precloacal supplements present or absent. Females didelphic-amphidelphic with reflexed ovaries. Tail often conical, sometimes with a cylindrical terminal portion.

**Type species:** *Desmodorella tenuispiculum* (*Allgén, 1928*) *Verschelde, Gourbault & Vincx, 1998*.

## Valid species of *Desmodorella Cobb, 1933*

The valid species list is based on *Verschelde, Gourbault & Vincx (1998)*, *Leduc & Zhao (2016)* and the *Nemys eds. (2025)*, with modifications:

*Desmodorella abyssorum* (*Allgén, 1929*) *Gerlach, 1963*
    Syn *Desmodora abyssorum Allgén, 1929*
*Desmodorella aquaedulcis* (*Gagarin & Nguyen, 2003*) *Decraemer & Smol, 2006*
    Syn *Desmodora aquaedulcis Gagarin & Nguyen, 2003*
*Desmodorella balteata Verschelde, Gourbault & Vincx, 1998*
*Desmodorella cornuta* **sp. nov.**
*Desmodorella curvispiculum* (*Jensen, 1985*) *Verschelde, Gourbault & Vincx, 1998*
    Syn *Desmodora curvispiculum Jensen, 1985*
*Desmodorella filispiculum* (*Lorenzen, 1976*) *Verschelde, Gourbault & Vincx, 1998*
    Syn *Desmodora filispiculum Lorenzen, 1976*
*Desmodorella papillostoma* (*Murphy, 1962*) *Verschelde, Gourbault & Vincx, 1998*
    Syn *Desmodora papillostoma Murphy, 1962*
*Desmodorella parabalteata* **sp. nov.**
*Desmodorella perforata* (*Wieser, 1954*) *Verschelde, Gourbault & Vincx, 1998*
    Syn *Desmodora perforata Wieser, 1954*
        *Desmodora wieseri Inglis, 1968*
        *Desmodora wolfgangi* (*Inglis, 1968*) *Gerlach & Riemann, 1973*
        *Xenodesmodora wieseri Inglis, 1968*
*Desmodorella sanguinea* (*Southern, 1914*) *Verschelde, Gourbault & Vincx, 1998*
    Syn *Desmodora sanguinea Southern, 1914*
*Desmodorella schulzi* (*Gerlach, 1950*) *Verschelde, Gourbault & Vincx, 1998*
    Syn *Desmodora schulzi Gerlach, 1950*
*Desmodorella sinuata* (*Lorenzen, 1976*) *Verschelde, Gourbault & Vincx, 1998*
    Syn *Desmodorella sinuata Lorenzen, 1976*
*Desmodorella spineacaudata Verschelde, Gourbault & Vincx, 1998*
*Desmodorella tenuispiculum* (*Allgén, 1928*) *Verschelde, Gourbault & Vincx, 1998*
    Syn *Desmodora (Desmodorella) cephalata Cobb, 1933*
        *Desmodora cephalata* (*Cobb, 1933*) *Gerlach & Riemann, 1973*
        *Desmodora cephalia* (*Cobb, 1933*) *Gerlach & Riemann, 1973*
        *Desmodora tenuispiculum Allgén, 1928*
        *Desmodorella cephalia* (*Cobb, 1933*) *Gerlach & Riemann, 1973*
*Desmodorella verscheldei Leduc & Zhao, 2016*

### Invalid species

*Desmodorella hirsuta* (*Chitwood, 1936*) *Verschelde, Gourbault & Vincx, 1998* (nomen dubium)

  *Desmodorella bullata* (*Steiner, 1916*) *Verschelde, Gourbault & Vincx, 1998* (taxon inquirendum)

## Description of the new species.

*Desmodorella cornuta* **sp. nov.**
(Table 2; Figs. 1–3)

**Material studied**. Holotype male (MOUFPE 0034), paratype female 1 (MOUFPE 0035), 2 male paratypes (NM-LMZOO/UFPE 511–512) and 2 female paratypes (NM-LMZOO/UFPE 513–514).

**Type locality**. South Atlantic Ocean, continental shelf off the State of Rio Grande do Norte (Potiguar Basin), Brazil; station ME2B2R3 (5°02′30.3″S, 36°23′12.3″W); June 2013; depth: 8.5 m.

**Locality of paratypes**. **Paratype female 1**: South Atlantic Ocean, continental shelf off the State of Rio Grande do Norte (Potiguar Basin), Brazil, (5°02′29.6″S, 36°23′11.9″W); June 2013; depth: 8.5 m. **Male paratypes**: (1) South Atlantic Ocean, continental shelf off the State of Rio Grande do Norte (Potiguar Basin), Brazil, (5°01′12.4″S, 36°23′27.6″W); June 2012; depth: 8.1 m; (2) South Atlantic Ocean, continental shelf off the State of Rio Grande do Norte (Potiguar Basin), Brazil, (05°02′29.6″S, 36°23′11.9″W); depth: 8.5 m. **Female paratypes 2 and 3**: South Atlantic Ocean, continental shelf off the State of Rio Grande do Norte (Potiguar Basin), Brazil, (5°01′12.4″S, 36°23′27.6″W); June 2012; depth: 8.1 m.

**Etymology**. The specific epithet "*cornuta*" refers to the presence of dorsally positioned, horn-shaped cuticular projections.

**Holotype male** (Figs. 1 and 2, Table 2). Body cylindrical (1,254 μm long), narrowest in region between base of the pharynx and anterior end of testis and widest posteriorly. Maximum body diameter corresponding to 2.2 times head diameter. Cuticle coarsely annulated and ornamented with transversal rows of small vacuoles. Cuticle pattern variable along body. Annules broad and widely spaced in anterior region (first 10 annules below head capsule = 30 μm), narrower and more closely in mid-body (10 annules = 18 μm); broader again from precloacal region to tail (10 = 20 μm). Twelve longitudinal rows of hair-like spines, sometimes difficult to see under light microscopy, arranged along body (5–7 μm), often indistinct under light microscope, most visible 85 μm from base of pharynx, extending to precloacal region. At 136 μm from pharynx base, two sublateral rows merge laterally, forming "false lateral alae" (cf. *Verschelde, Gourbault & Vincx, 1998*) of shorter, more robust spines, extending to testis region. Afterward, rows diverge, and spines morphology returns to regular form. Somatic setae (3–8 μm) arranged in two longitudinal rows (dorsal and ventral) along body, absent in tail region. Dorsal, protuberant horn-shaped cuticular projection (nine μm long) located dorsally at the 14th annule (64.5 μm from

**Table 2  Morphometric data of *Desmodorella cornuta* sp. nov.** The measurements are expressed in micrometers, or if noted, as percentages or ratios. Not applicable (*); not available for measurement (-); a, b, c, c' = de Man's ratios (*1880*).

| *Desmodorella cornuta* sp. nov. | Holotype (Male) | Male paratypes (*n* = 2) | Paratype (Female 1) | Other female paratypes (*n* = 2) |
|---|---|---|---|---|
| Body length | 1,254 | 1,107–1,092 | 1,221 | 1,014–1,074 |
| Outer labial setae length | 2 | 2–2.5 | – | – |
| Cephalic setae length | 3.5 | 3–4 | 3 | 3–4 |
| Head diameter at level of the cephalic setae | 21.5 | 20.5–22 | 23.5 | 18.5–23 |
| Cephalic setae in relation to head diameter at the cephalic setae level (%) | 16% | 14–19.5% | 13% | 16%–17% |
| Distance from anterior end to amphidial fovea | 3.5 | 6–6.5 | – | 2.5–4 |
| Amphidial fovea diameter (maximum width) | 11.5 | 11–11.5 | 11 | 11 |
| Body diameter at level of the amphidial fovea | 26.5 | 26–26.5 | 25.5 | 21.5–25.5 |
| % of the amphidial fovea diameter in relation to corresponding body diameter | 43% | 42–50% | 43% | 43–51% |
| Pharynx length | 145 | 142–143 | 142 | 137–138 |
| Distance between the horn-shaped cuticular projection to anterior end | 64,5 | 54.5–61.5 | 56 | 53.5–56 |
| Length of horn-shaped cuticular projection | 9 | 8.5–9 | 8 | 9 |
| Position of the horn-shaped cuticular projection in relation to the pharynx length (%) | 44% | 38–43% | 39% | 39%–41% |
| Pharyngeal bulb diameter | 17 | 19 | 13 | 19 |
| Body diameter at level of the pharyngeal bulb | 34.5 | 31 | 34.5 | 30–32 |
| % of basal bulb diameter in relation to corresponding body diameter | 49% | 61% | 38% | 59–63% |
| Body diameter at the level of the pharynx end | 35 | 29–30.5 | 35 | 30–31 |
| Maximum body diameter | 48 | 34–55 | 57 | 42–44.5 |
| Anal or cloacal body diameter | 30 | 26–27 | 25.5 | 24–25 |
| Tail length | 118 | 109.5–114 | 108.5 | 93–106 |
| Length of spicules along arc | 79 | 55–71.5 | * | * |
| Length of spicules along cord | 41 | 50–64 | * | * |
| Length of gubernaculum | 17 | 17 | * | * |
| Length of gubernaculum in relation to length of spicules along arc (%) | 21.5% | 24% | * | * |
| Length of spicules along arc in relation to cloacal body diameter | 2.6 | 2–2.6 | * | * |
| Distance from anterior end to vulva | * | * | 858 | 750–756 |
| Position of vulva from anterior end (%) | * | * | 70% | 70–75% |
| Body diameter in vulva region | * | * | 57 | 42–44.5 |
| Anterior ovary length | * | * | 81 | 155.5–159 |
| Posterior ovary length | * | * | 77.5 | 106–108 |
| Reproductive system length | 385.5 | 273 | 98.5 | 130–141 |
| % of reproductive system in relation to body length | 31% | 25% | 8% | 12–14% |
| a | 26 | 20–32 | 21 | 24 |
| b | 8.7 | 7.7 | 8.6 | 7.4–7.8 |
| c | 10.6 | 9.6–10 | 11.3 | 9.6–11.6 |
| c' | 4 | 4–4.4 | 4.3 | 4.4–3.7 |

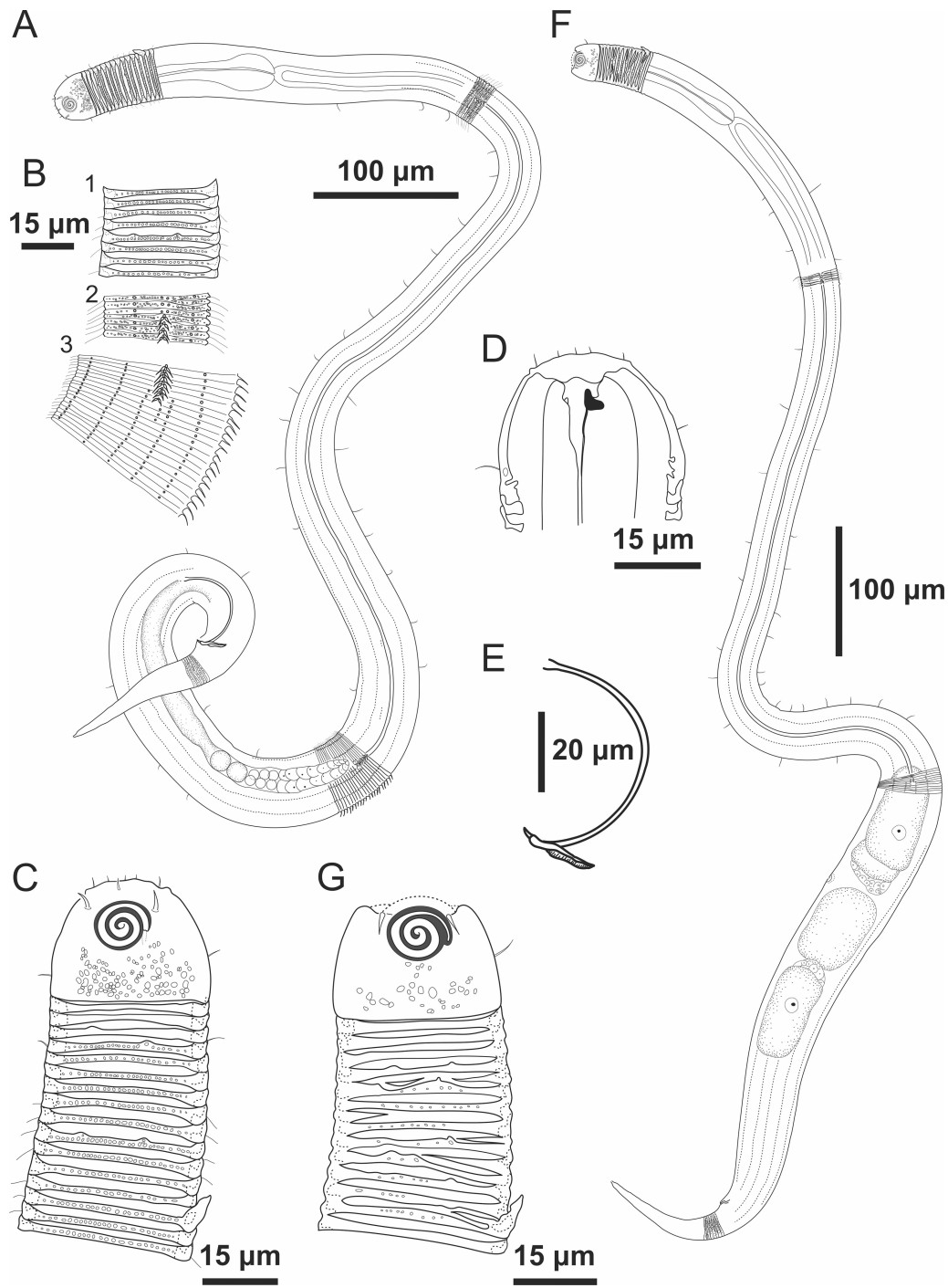

**Figure 1** ***Desmodorella cornuta* sp. nov. holotype male and paratype female 1.** Holotype male and paratype female 1. Holotype male: (A) whole body overview; (B) cuticle details - 1: at the pharynx level; 2: at the beginning of the false lateral alae; 3: at the end of the of the false lateral alae, (C) anterior region, (D) buccal cavity, (E) spicule and gubernaculum. Paratype female 1: (F) whole body overview, (G) anterior region.

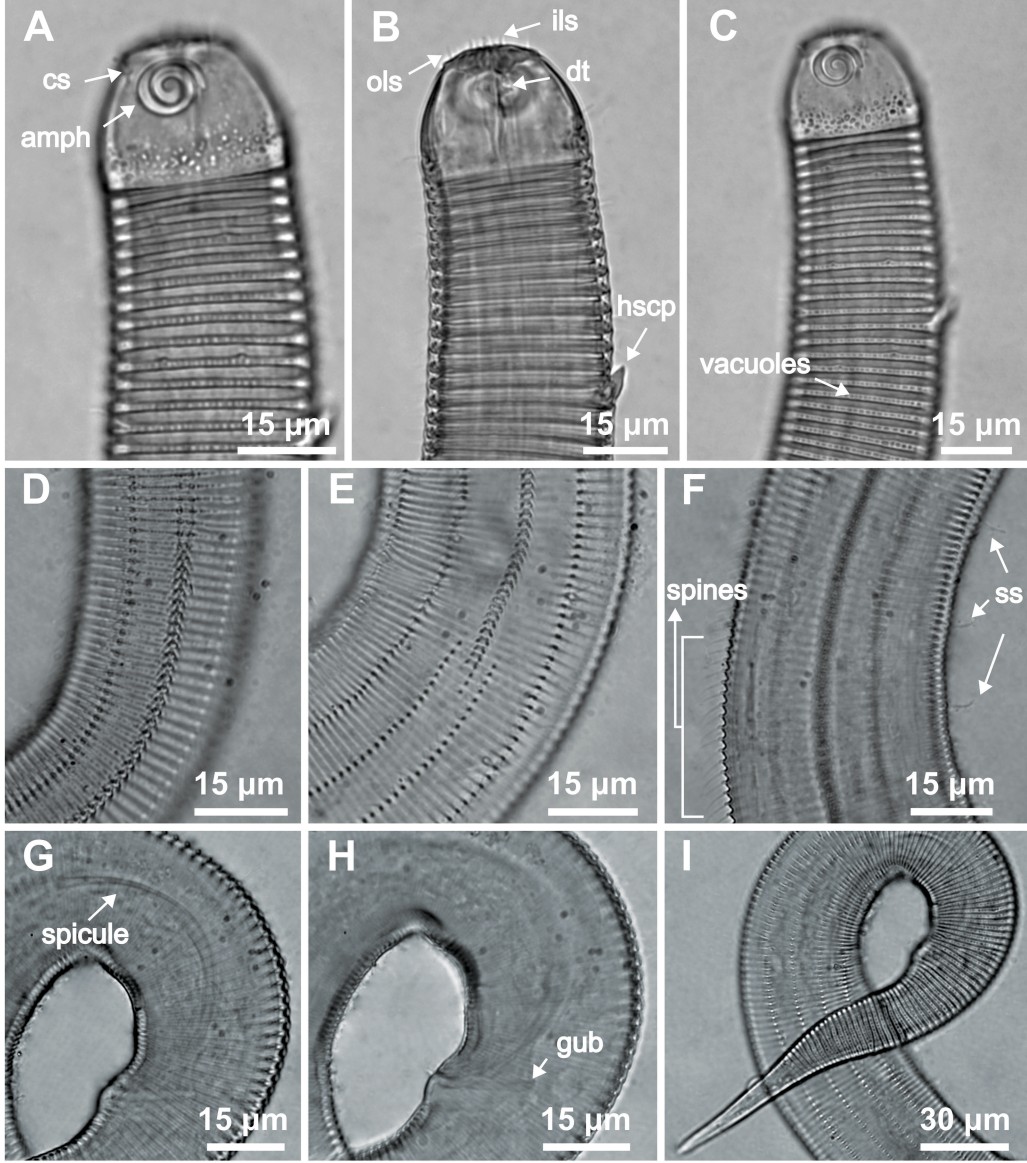

**Figure 2** *Desmodorella cornuta* **sp. nov. holotype male.** (A) Anterior end (cs: cephalic setae; amph: amphidial fovea), (B) anterior end (ils: inner labial setae; ols: outer labial setae; dt: dorsal tooth; hscp: horn-shaped cuticula projection), (C) anterior region, (D) beginning of the false lateral alae, (E) end of the of the false lateral alae, (F) cuticular hair-like spine and somatic setae (ss), (G) spicule, (H) gubernaculum (gub), (I) tail.

anterior end, or 44% of pharynx length). Head capsule long, well-developed, ornamented with numerous small vacuoles below amphidial fovea. Anterior sensilla arranged in 6+6+4 pattern: six inner labial papillae, six outer labial papillae (about two μm long) and four small cephalic setae (3.5 μm long). Cephalic setae corresponding to 16% of head diameter. Rows of subcephalic setae absent. Two additional setae (about three μm long), one dorsal and one ventral, present on posterior part of head capsule. Amphidial fovea distinctly

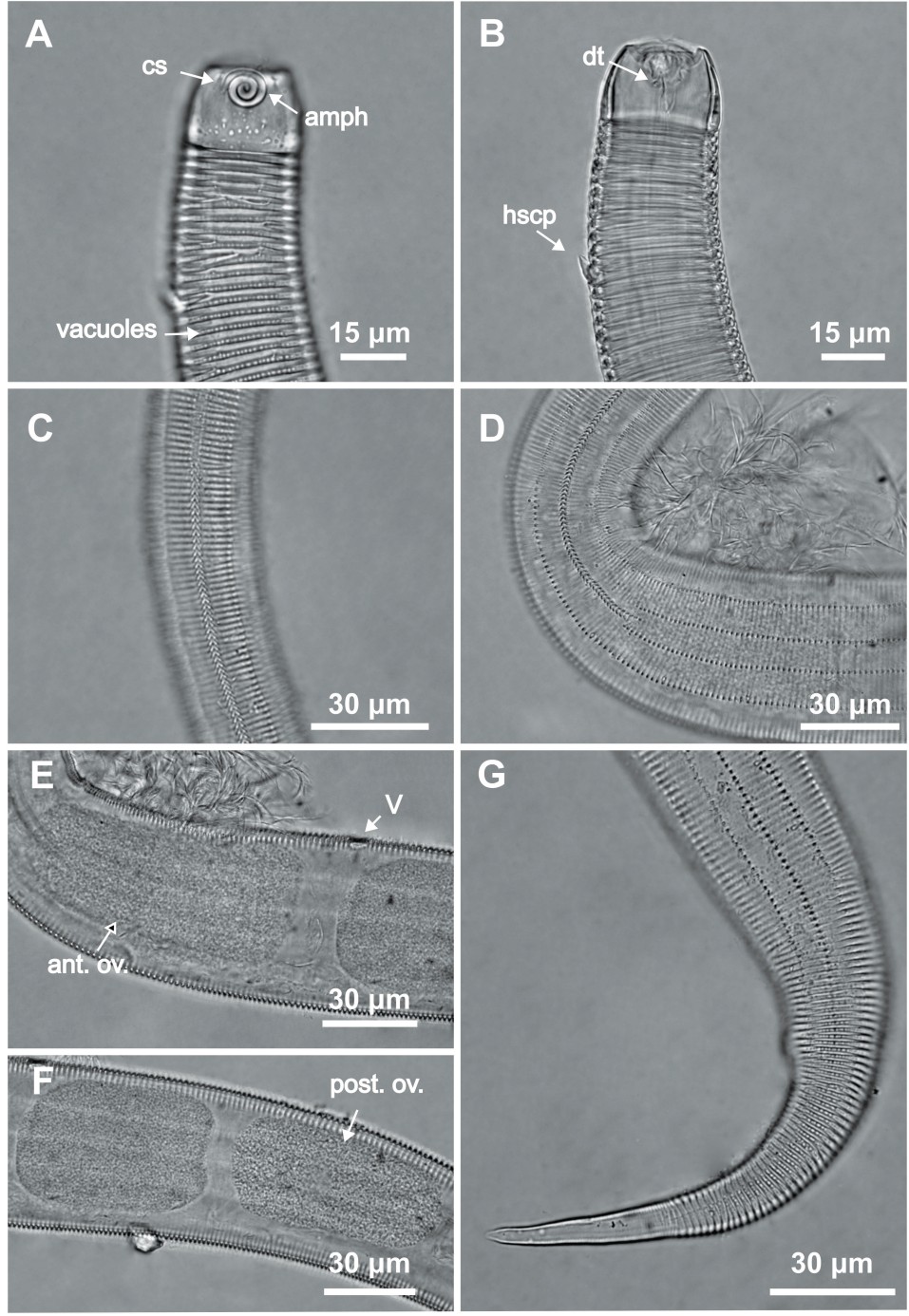

**Figure 3** ***Desmodorella cornuta*** **sp. nov. paratype female 1.** (A) Anterior region (cs: cephalic setae; amph: amphidial fovea); (B) anterior region (dt: dorsal tooth; hscp: horn-shaped cuticular projection), (C) beginning of the false lateral alae; (D) end of the of the false lateral alae, (E) reproductive system (V: vulva; ant.ov.: anterior ovary), (F) posterior ovary (post. ov.), (G) tail.

sclerotized, multispiral, about three turns, 43% of corresponding body diameter and located 3.5 μm from anterior end (about 0.2 times the head diameter). Buccal cavity with a strong dorsal tooth and a small ventrosublateral tooth. Pharynx muscular (145 μm long), cylindrical forming slightly oval terminal bulb that occupies 49% of corresponding body diameter. Nerve ring, secretory-excretory system and cardia not observed. Reproductive system monorchic, with single anterior outstretched testis on left of intestine. Spicules slender (79 μm long), arched ventrally, with slightly swollen proximal end (2.6 times cloacal body diameter) and without capitulum. Gubernaculum with short lateral crurae. Precloacal supplements absent. Caudal glands indistinct. Tail conical, elongated, about 4 times cloacal body diameter.

**Paratype female 1** (**Figs. 1** and **3**, **Table 2**). Generally similar to male. Body 1,221 μm long, maximum diameter of 57 μm at vulva level (about 2.4 times head diameter). Cuticular annule pattern as in male (first 10 annules below head capsule = 27 μm; 10 annules in narrowest body region = 16 μm; 10 annules in tail = 19 μm). Several incomplete or bifurcated annules present, more visible at pharynx level. Longitudinal rows of hair-like spines and head capsule similar to male. Somatic setae as in male visible along anterior two-thirds of the body. Dorsal horn-shaped cuticular projection (8 μm long) positioned dorsally at 13th annule (56 μm from anterior end or 39% of pharynx length). Labial region invaginated. Cephalic setae correspond to 13% of head diameter. Amphidial fovea as in male. Basal bulb occupies 38% of corresponding body diameter. Vulva located 858 μm from anterior end, at 70% of body length. Reproductive system didelphic with reflexed ovaries. Tail conical, elongated, 4.3 times anal body diameter.

**Diagnosis**. *Desmodorella cornuta* **sp. nov.** is characterized by the following combination of the features: cuticle coarsely annulated and ornamented with transversal rows of small vacuoles; protuberant horn-shaped cuticular projection located dorsally at 38–44% of pharynx length; twelve longitudinal rows of hair-like spines arranged along of body; two pairs of lateral rows with more distinct spines forming "false lateral alae"; head capsule ornamented with numerous small vacuoles; amphidial fovea multispiral (about 3 turns), occupying 42–51% of corresponding body diameter; subcephalic setae absent with additional setae present; tail conical (3.7–4.4 times cloacal/anal body diameter); males with slender, ventrally arched spicules (55–79 μm long; 2–2.6 times cloacal body diameter), with slightly swollen proximal end.

**Differential diagnosis** (**Table 3**). *Desmodorella cornuta* **sp. nov.** shares with *D. curvispiculum* (*Jensen, 1985*) *Verschelde, Gourbault & Vincx, 1998*, *D. perforata* (*Wieser, 1954*) *Verschelde, Gourbault & Vincx, 1998* and *D. balteata Verschelde, Gourbault & Vincx, 1998*, the following features: head capsule ornamented with numerous small vacuoles, similar spicules length (see Table 3), and absence of subcephalic setae. Additionally, *D. cornuta* **sp. nov.** and *D. balteata* possess "false lateral alae". However, *D. cornuta* **sp. nov.** is the only known species of *Desmodorella* that exhibits a protuberant, horn-shaped cuticular projection dorsally in the pharynx region (between 38–44% of pharynx length). This unique feature aids in the identification of *D. cornuta* **sp. nov.** and clearly distinguishes it from *D. curvispiculum*, *D. perforata*, *D. balteata* and the other valid species of the genus.

**Table 3 Comparison of species *Desmodorella cornuta* sp. nov. with morphologically similar species.**
a, b, c, de Man's ratios (*1880*); parameter absent (-); parameter present (+). An asterisk (*) indicates two pairs of lateral rows of more distinct spines, among the other rows of spines (referred to by *Verschelde, Gourbault & Vincx (1998)* as "false lateral alae"). Two asterisks (**) indicate protuberant horn-shaped cuticular projection positioned dorsally in the pharyngeal region.

| | *Desmodorella curvispiculum* | *D. perforata* | *D. balteata* | *D. cornuta* sp. nov. |
|---|---|---|---|---|
| Body length (µm) | 1,004–1,042 | 1,850–1,410 | 867–1,078 | 1,014–1,254 |
| a | 17–26 | 28.6–32.4 | 15.7–26.3 | 32–20 |
| b | 7.3–7.6 | 7.4–8.8 | 5.9–7.6 | 7.4–8.7 |
| c | 10.9–11.5 | 11.6–13.5 | 8.7–12.3 | 9.6–11.6 |
| Spicules length (µm) | 76 | 52 | 85–65 | 55–79 |
| False lateral alae* | - | - | + | + |
| Subcephalic setae | - | - | - | - |
| Horn-shaped cp.** | - | - | - | + |

*Desmodorella parabalteata* **sp. nov.**

(Table 4; Figs. 4–8)

**Material studied**. Holotype male (MOUFPE 0036), paratype female 1 (MOUFPE 0037), 10 male paratypes (NM LMZOO-UFPE 515–524) and eight female paratypes (NM LMZOO-UFPE 525–532).

**Type locality**. South Atlantic Ocean, continental shelf of the State of Bahia, Brazil (13°04′10.32″S, 38°25′46.98″W); 11 December 2019; depth: 65 m.

**Locality of paratypes**. **Paratype female 1**: South Atlantic Ocean, continental shelf of the State of Sergipe, Brazil, (11°00′00.54″S, 36°49′58.98″W); 10 December, 2019; depth: 54 m. **Male paratypes**: (1 and 2) Same as holotype locality, Bahia, Brazil, 11 December 2019; depth: 65 m; (3–7) Sergipe, Brazil, (11°00′00.54″S, 36°49′58.98″W), 10 December 2019; depth: 54 m; (8) Sergipe, Brazil, (10°44′59.28″S, 36°25′32.88″W); 09 December 2019; depth: 58 m; (9 and 10) Alagoas, Brazil, (10°07′05.7″S, 35°50′58.0″W); 09 December 2019; depth: 63 m. **Other female paratypes**: (2, 3, 7) Sergipe, Brazil, (11°00′00.54″S, 36°49′58.98″W); 10 December 2019; depth: 54 m; (4, 5, 9) Bahia, Brazil, (13°04′10.32″S, 38°25′46.98″W), 11 December 2019; depth: 65 m; (6, 8) Alagoas, Brazil, (10°07′05.7″S, 35°50′58.0″W); 09 December 2019; depth: 63 m.

**Etymology**. The specific epithet "*parabalteata*" refers to the morphological similarity of this species to *Desmodorella balteata*.

**Holotype male (Figs. 4 and 5, Table 4)**. Body cylindrical, 697.5 µm long; narrowest between base of pharynx and anterior end of testis; widest at level of testis. Maximum body diameter corresponding to 2.1 times head diameter. Cuticle annulated, ornamented with transversal rows of small vacuoles (more evident in pharyngeal region). Cuticle pattern variable along body. Annules broad in anterior pharyngeal region (first 10 annules below head capsule = 16.5 µm), gradually narrowing towards widest body (10 annules = 6 µm) expanding progressively from proximal region of spicules to tail (10 annules = 11 µm). Longitudinal rows of ridges or short spines indistinct in holotype under light

**Table 4 Morphometric data of *Desmodorella parabalteata* sp. nov. The measurements are expressed in micrometers, or if noted, as percentages or ratios.** Not applicable (*); not available for measurement (-); a, b, c, c' = de Man's ratios (*1880*).

| *Desmodorella parabalteata* sp. nov. | Holotype (Male) | Male paratypes ($n = 10$) | Paratype (Female 1) | Other female paratypes ($n = 9$) |
|---|---|---|---|---|
| Body length | 697.5 | 685.5–817.5 | 795 | 583–814.5 |
| Outer labial setae length | 3 | 2 | 2.5 | 2 |
| Cephalic setae length | 4 | 4 | 4 | 4 |
| Head diameter at level of the cephalic setae | 19.5 | – | – | 16–17 |
| Cephalic setae in relation to head diameter at the cephalic setae level (%) | 21% | – | – | 24–25% |
| Distance from anterior end to amphidial fovea | 6.5 | 5–7 | – | 5–8 |
| Amphidial fovea diameter (maximum width) | 11.5 | 10–11.5 | 11 | 9–12 |
| Body diameter at level of the amphidial fovea | 22 | 20–23 | 21.5 | 19–23 |
| % of the amphidial fovea diameter in relation to corresponding body diameter | 52% | 43–55% | 51% | 43–59% |
| Pharynx length | 123 | 113.5–133.5 | 131.5 | 109–129 |
| Pharyngeal bulb diameter | 18 | 19–25.5 | 21.5 | 17–23 |
| Body diameter at level of the pharyngeal bulb | 32.5 | 30.5–36 | 31 | 28–32.5 |
| % of basal bulb diameter in relation to corresponding body diameter | 55% | 62–73% | 70% | 56–75% |
| Body diameter at the level of the pharynx end | 29 | 24.5–34 | 30 | 25–32 |
| Maximum body diameter | 40.5 | 37–51 | 51 | 39–51 |
| Anal or cloacal body diameter | 21 | 17.5–24.5 | 23 | 19–23 |
| Tail length | 113 | 88–113.5 | 99 | 72–107 |
| Length of the non-annulated tail end | 13 | 13–17 | 15.5 | 9–23.5 |
| Length of spicules | 36.5 | 25–41.5 | * | * |
| Length of gubernaculum | 14.5 | 12–18 | * | * |
| Length of gubernaculum in relation to length of spicules (%) | 40% | 37–56% | * | * |
| Length of spicules along arc in relation to cloacal body diameter | 1.7 | 1.4–1.8 | * | * |
| Distance from anterior end to vulva | * | * | 510 | 389.5–525 |
| Position of vulva from anterior end (%) | * | * | 64% | 64–68% |
| Body diameter in vulva region | * | * | 51 | 39–51 |
| Anterior ovary length | * | * | 148 | 108–270 |
| Posterior ovary length | * | * | 130 | 97–271.5 |
| Reproductive system length | 232.5 | 217–317 | 210 | 142.5–229.5 |
| % of reproductive system in relation to body length | 33% | 27–39% | 26% | 23–28% |
| a | 17 | 14–21 | 16 | 13–17 |
| b | 6 | 6–6.5 | 6 | 5–6 |
| c | 6 | 7–8 | 8 | 7–9 |
| c' | 5 | 4–6 | 4 | 4–5 |

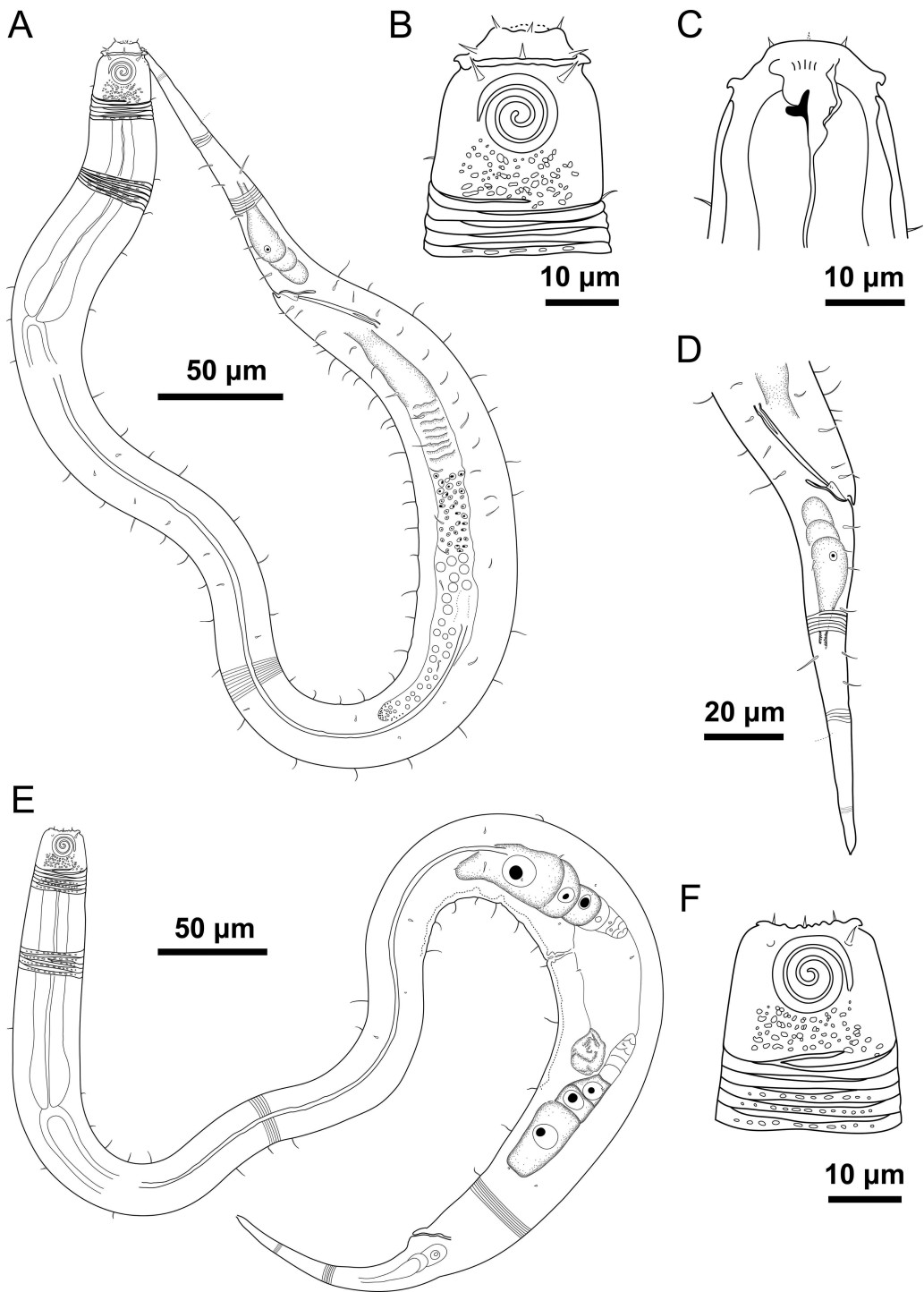

**Figure 4** *Desmodorella parabalteata* **sp. nov. holotype male and paratype female 1.** Holotype male: (A) whole body overview; (B) anterior end (C) buccal cavity, (D) posterior end. Paratype female 1: (E) whole body overview, (F) anterior end.

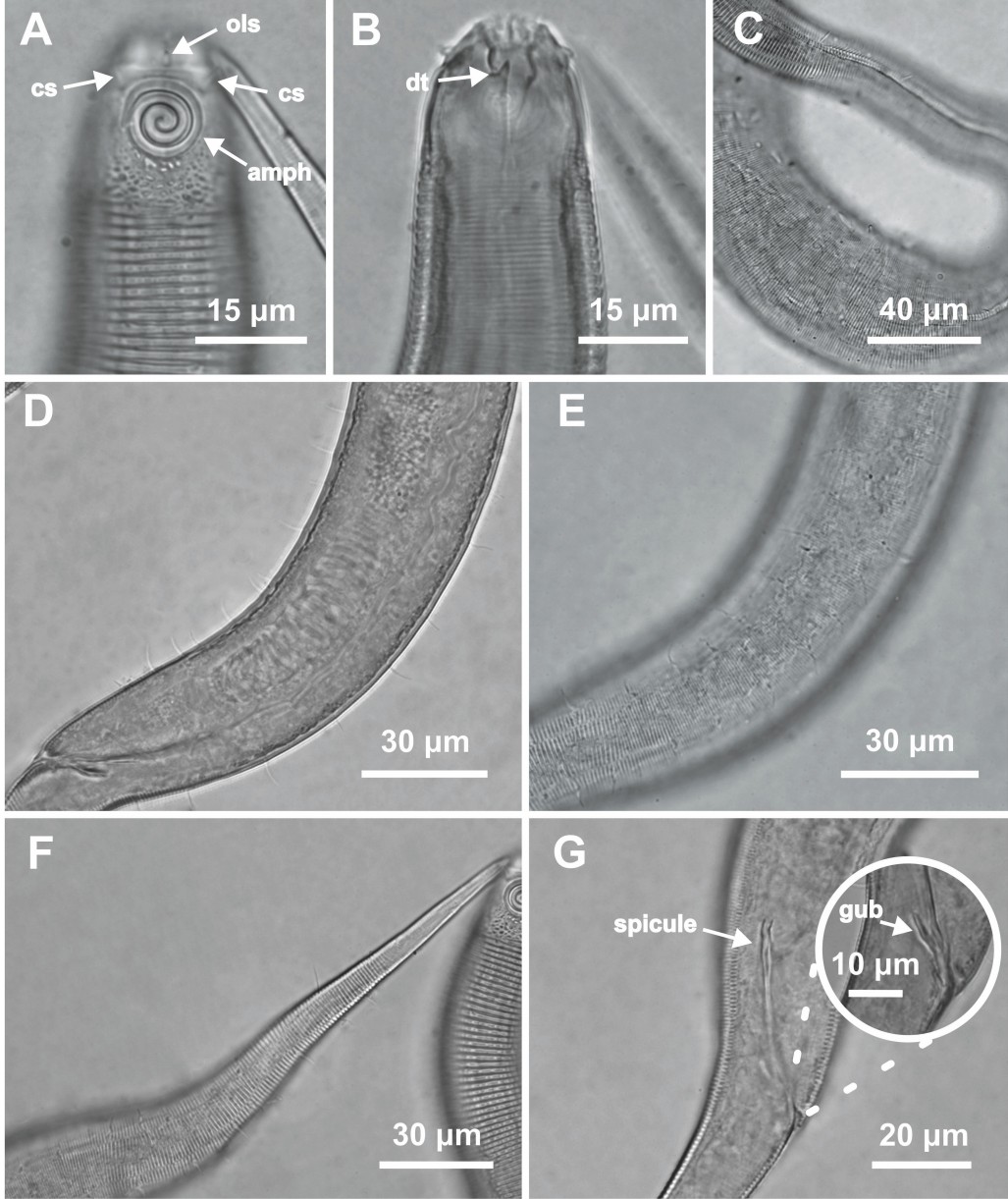

**Figure 5** *Desmodorella parabalteata* **sp. nov. holotype male and male paratype 9.** Holotype male: (A) anterior end (ols: outer labial setae; cs: cephalic setae; amph: amphidial fovea), (B) anterior end (dt: dorsal tooth), (D and E) rows of somatic setae, (F) tail, (G) spicule (spic) and gubernaculum (gub). Male paratype 9: (C) longitudinal rows of ridges.

microscopy, but visible in paratype male 9 (Fig. 5C). Two sublateral pairs of longitudinal spines converge laterally about 30 µm from base of pharynx, forming ''false lateral alae'' composed of more robust spines, extending to first third of testis. Somatic setae arranged in six longitudinal rows (four sublateral, one dorsal, and one ventral). Dorsal and ventral rows of somatic setae 2–7 µm long, present along entire body except tail region. Sublateral somatic setae <2–6 µm long, extend from 70 µm behind pharynx to caudal region; smaller

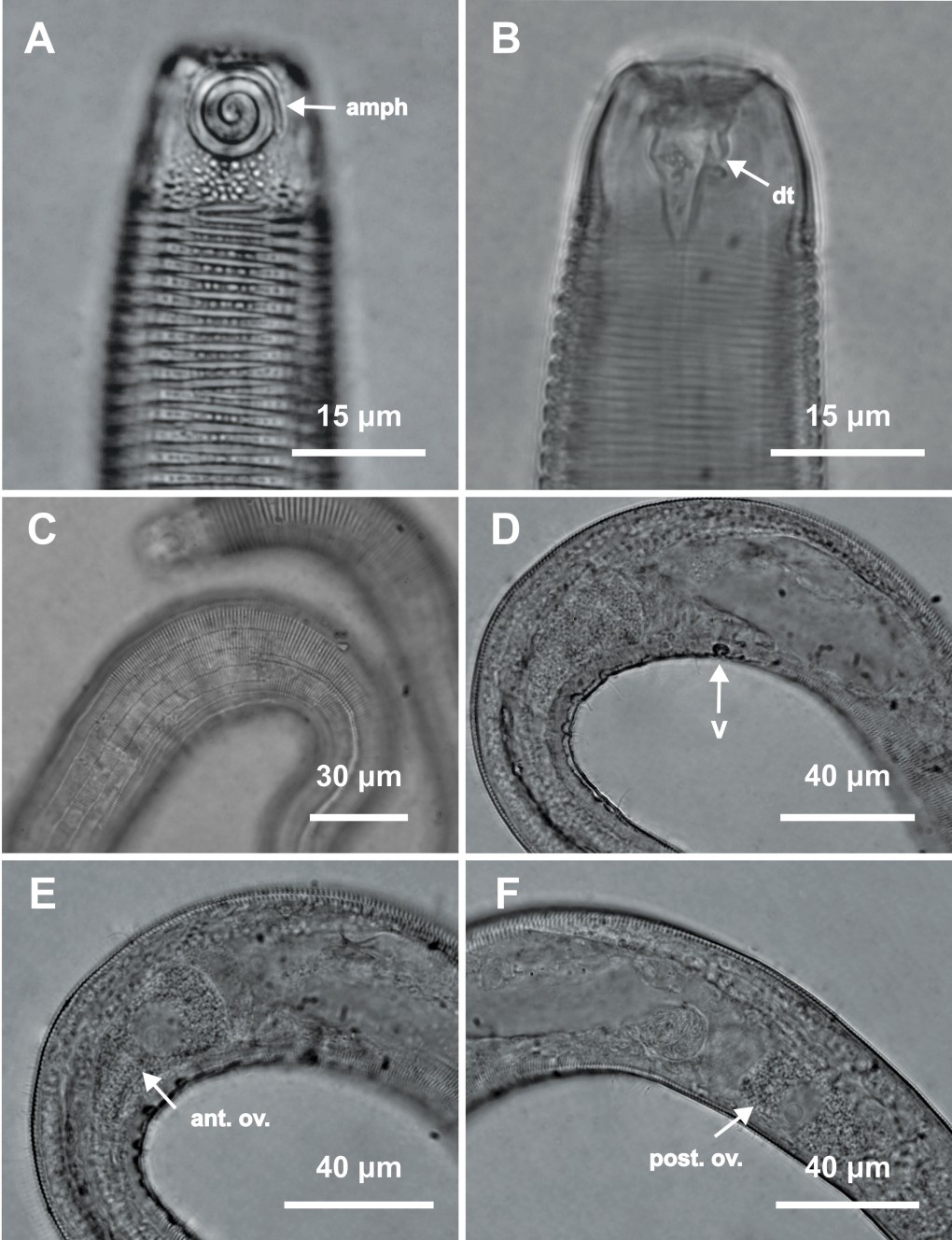

**Figure 6** *Desmodorella parabalteata* **sp. nov. female paratype 1 and female paratype 9.** Female paratype 1: (A) anterior end (amph: amphidial fovea), (B) buccal cavity (dt: dorsal tooth), (D) vulva region (V: vulva), (E) anterior ovary (ant.ov.), (F) posterior ovary (post. ov.). Female paratype 9: (C) longitudinal rows of ridges.

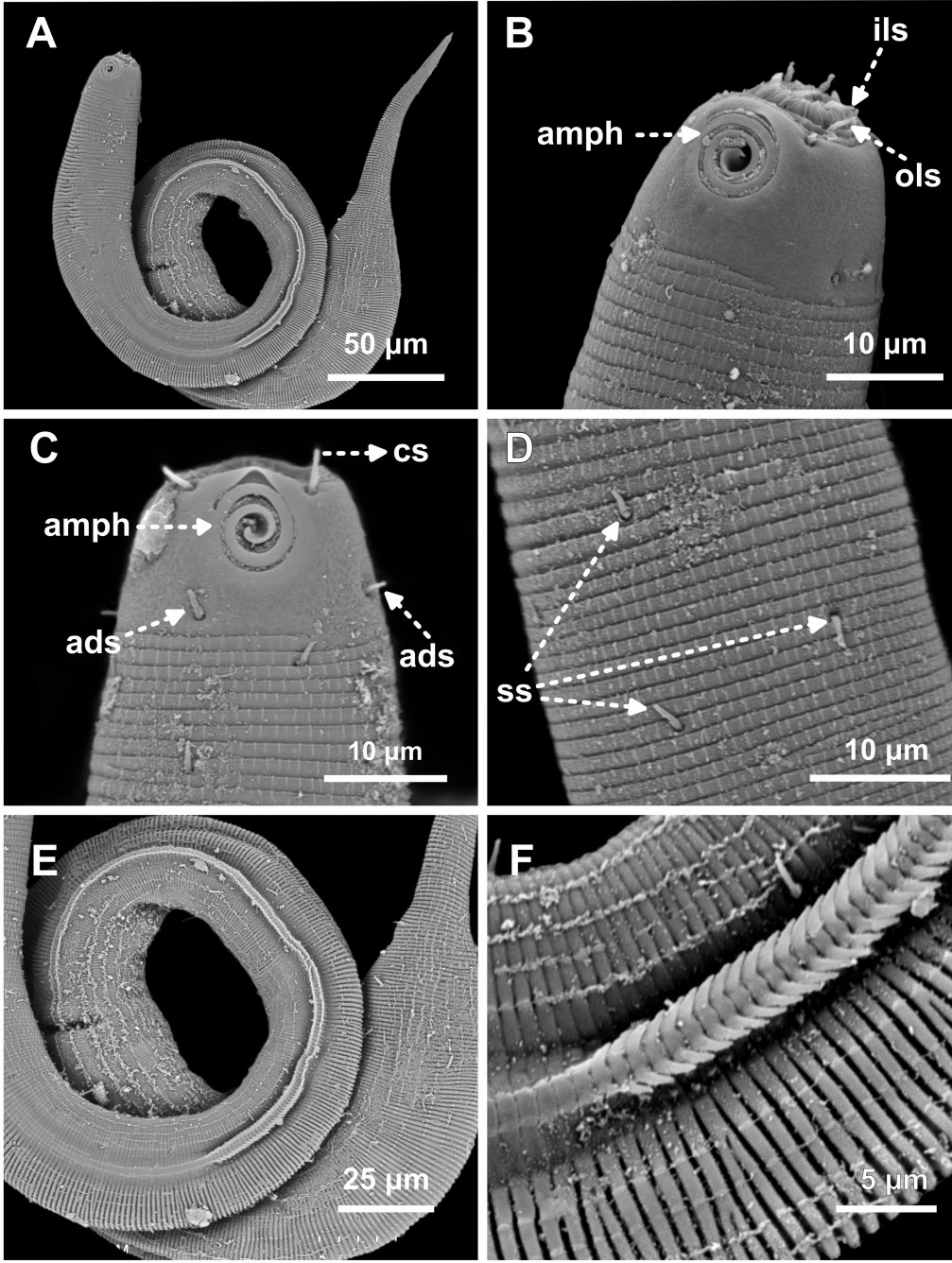

**Figure 7** *Desmodorella parabalteata* **sp. nov. male paratypes 8 and 10, SEM photographs.** Paratype male 10: (A) whole body overview; (B) anterior end (ils: inner labial setae; ols: outer labial setae; amph: amphidial fovea); (E) false lateral alae and longitudinal rows of ridges; (F) beginning of the false lateral alae. Paratype male 8: (C) anterior end (cs: cephalic setae; amph: amphidial fovea; ads: additional setae); (D) cuticular ornamentation at the pharynx level (ss: somatic setae).

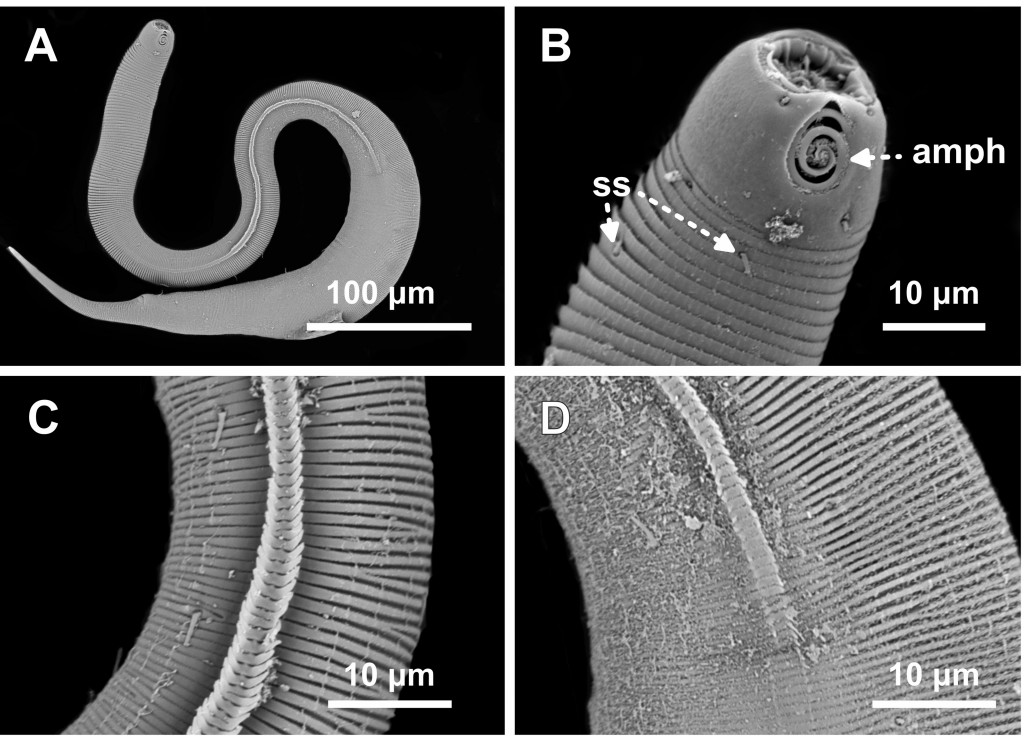

**Figure 8** *Desmodorella parabalteata* **sp. nov. paratype female 3, SEM photographs.** (A) Whole body overview; (B) anterior end (amph: amphidial fovea; ss: somatic setae); (C) false lateral alae and longitudinal rows of ridges; (D) posterior end of the false lateral alae.

and less visible in narrowest region; longer and slightly more robust near precloacal region. Head capsule long, well-developed, ornamented with numerous small vacuoles below amphidial fovea. Anterior sensilla arranged in 6 + 6 + 4 pattern: six inner labial papillae, six outer labial setae (about three μm long) and four small cephalic setae (four μm long). Cephalic setae corresponding to 21% of head diameter. Rows of subcephalic setae absent. Two additional setae, one dorsal and other ventral (about two μm long). Amphidial fovea distinctly sclerotized, multispiral, about 3.5 turns, occupying 52% of body diameter; anterior edge aligned with cephalic setae and located 6.5 μm from anterior end (about 0.3 times head diameter). Buccal cavity with a strong dorsal tooth and a small ventrosublateral tooth. Pharynx muscular 123 μm long, cylindrical with slightly oval terminal bulb (55% of corresponding body diameter). Nerve ring, secretory-excretory system, and cardia not observed. Reproductive system monorchic with anterior outstretched testis on left of intestine. Spicules slender, 36.5 μm long (1.7 times cloacal body diameter), nearly straight, with slightly swollen proximal end, without capitulum. Gubernaculum funnel-shaped, surrounding distal end of spicules. Precloacal supplements absent. Three caudal glands. Tail conical, elongated, about 5 times cloacal body diameter.

**Paratype female 1 (Figs. 4 and 6, Table 4).** Largely similar to male. Body 795 μm long; maximum diameter 51 μm at vulva level. Cuticular annule similar to male (first 10 annules below head capsule = 20 μm; 10 annules in widest body region= 6.5 μm; 10 annules

in tail = 13 µm). Longitudinal rows of ridges visible under light microcopy in paratype females 6, 7 and 9 (Fig. 6C); indistinct in paratype female 1. Somatic setae (<2–8 µm long) arranged similarly to male, but fewer in number; present along entire body except tail region. Head capsule largely similar to male. Labial region invaginated. Cephalic setae correspond to 22% of head diameter. Amphidial fovea similar to male. Basal bulb occupies 70% of corresponding body diameter. Vulva 510 µm from anterior end, located at 64% of body length. Reproductive system didelphic with reflexed ovaries. Three caudal glands present. Tail conical, elongated, 4 times anal body diameter.

**SEM analyses**. **Male paratypes 8 and 10 (Fig. 7) and paratype female 3 (Fig. 8):** Head capsule wrinkled (Figs. 7B, 7C, 8B). Cuticular annules in anterior third of body with numerous transverse bars (Figs. 7B, 7C, 7D, 8B). Towards mid-body (narrowest region), bars number decreases and bars elongate forming short spine-like structures arranged in longitudinal rows (Fig. 7E). Exact number of longitudinal rows indeterminate. Bifurcated cuticular annules occur along body. Spines of false lateral alae largest at anterior end of rows, progressively decreasing in size posteriorly (Figs. 7E and 8C). After termination of false lateral alae, rows diverge and spines resume morphology seen in other rows (Fig. 8D).

**Diagnosis**. *Desmodorella parabalteata* **sp. nov.** is characterized by a combination of the following features: cuticle with coarse annulations and ornamented with transversal rows of small vacuoles; cuticle with numerous transverse bars under SEM; longitudinal rows of ridges or short spines (often indistinct under light microscopy) present along body; two pairs of lateral rows with more distinct spines among other rows (false lateral alae); somatic setae arranged in six longitudinal rows (four sublateral, one dorsal, and one ventral); head capsule ornamented with numerous small vacuoles below amphidial fovea; head capsule wrinkled under SEM; amphidial fovea multispiral, about 3.5 turns, with anterior edge at same level as cephalic setae and occupying 43–59% of corresponding body diameter; subcephalic setae absent; additional setae present; tail elongate-conical (4–6 times cloacal/anal body diameter); males with slender, nearly straight spicules (25–41.5 µm long; 1.4–1.8 times cloacal body diameter), slightly swollen proximally, without capitulum.

**Differential diagnosis.** *Desmodorella parabalteata* **sp. nov.** is closely related to *D. balteata*. These two species share several morphological features, including: a cephalic capsule ornamented with numerous small vacuoles below the amphidial fovea; absence of subcephalic setae; multispiral amphidial fovea; longitudinal rows of ridges often indistinct under light microcopy; two pairs of lateral rows with more distinct spines among other rows (false lateral alae) and six longitudinal rows of somatic setae. Together, these traits distinguish both species from other members of the genus *Desmodorella*.

However, *D. parabalteata* **sp. nov.** differs from *D. balteata* in the following features: presence of cuticular vacuoles in *D. parabalteata* **sp. nov.** (*versus* (*vs*) absent in *D. balteata*; number of amphidial turns (3.5 turns in the new species *vs* 2.6 turns in *D. balteata*); spicules length (25–41.5 µm long in *D. parabalteata* **sp. nov.** *vs* 65–85 µm long in *D. balteata*) and spicules morphology (slender, nearly straight spicules with slightly swollen proximal ends and lacking a capitulum in *D. parabalteata* **sp. nov.** *vs* slightly curved spicules with a small rounded capitulum in *D. balteata*). Additionally, males of *D. balteata* possess a ventral row of robust precloacal setae, a feature absent in the new species.

## Dichotomous identification key for valid species of *Desmodorella Cobb, 1933*

1. Amphidial fovea multispiral (2 or more turns)...............................................................2

- Amphidial fovea multispiral (less than 2 turns) or not multispiral...................................3

2. Spicules longer than 100 µm…............................................................................................4

- Spicules shorter than 100 µm…...........................................................................................5

3. Spicules longer than 100 µm…………………………………………………………..6

- Spicules shorter than 100 µm…...........................................................................................9

4. Amphidial fovea completely positioned in the main part of the head capsule; vacuolated head capsule present…………...……...................................................................................7

- Amphidial fovea with anterior edge in lip region and posterior edge in head capsule; smooth head capsule; four ventrosublateral rows of 3–4 thorns on tail………………………………………………………..………*D. spineacaudata*

5. Head capsule not medially bulging; gubernaculum less than $^1/_3$ of spicules length......................................................................................................................8

- Head capsule strongly bulging at the level of amphidial fovea; gubernaculum equivalent to about $^1/_2$ of spicules length……………………………….……....…….*D. abyssorum*

6. Amphidial fovea cryptospiral; "false lateral alae" absent…………………………..……………………………………………….…11

- Amphidial fovea loop-shaped; "false lateral alae" present……………....*D. verscheldei*

7. Spicules 220–250 µm, sinuous posteriorly; short subcephalic setae; "false lateral alae" present; 12 longitudinal rows of hair-like spines.........................................................*D. sinuata*

- Spicules ~about 150 µm, arched; elongated cephalic and subcephalic setae; "false lateral alae" absent.................................................................................................*D. papillostoma*

8. "False lateral alae" present (2 pairs of lateral rows of more distinct spines)..………………………………………………………………..………..12

- "False lateral alae" absent……….…………………………..………..…..………………...13

9. Amphidial fovea loop-shaped (spiral)…………..……………………….………….10

- Amphidial fovea unispiral…………………………...…......……………..*D. aquaedulcis*

10. Elongated subcephalic setae (2 circles with 8 subcephalic setae each); smooth cephalic capsule; 8–12 longitudinal rows of hair-like spines; precloacal supplements as 4 rows of triangular cuticular spines; "false lateral alae" present; shoe-shaped gubernaculum…………………….………………….............................. ..*D. schulzi*\*

- Short subcephalic setae; vacuolated head capsule posterior to amphids; lamellar gubernaculum……………………………………………………………..*D. perforata*

11. Spicules 240–325 µm; 16 longitudinal rows of hair-like spines………..*D. filispiculum*

- Spicules 182–224 µm; 10 longitudinal rows of hair-like spines
……………………………………………………...…………………*D. sanguinea*

12. Horn-shaped cuticular projection dorsally at pharynx level; 2 longitudinal rows of somatic setae………………………………………………………….*D. cornuta* **sp. nov.**

- Horn-shaped cuticular projection absent; 6 longitudinal rows of somatic setae…………………………………………….................................................14

13. Subcephalic setae absent; 10–14 longitudinal rows of ridges..………*D. curvispiculum*

- Subcephalic setae present; 12–24 longitudinal rows of ridges……...…*D. tenuispiculum* \*\*

14. Cuticular vacuoles absent; spicules 65–85 µm, slightly arched with a tiny rounded capitulum; ventral row of robust precloacal setae present……………………………………………………..………………*D. balteata*

- Cuticular vacuoles present; spicules 25–41.5 µm, nearly straight with slightly swollen proximal end, no capitulum……………….………………..*D. parabalteata* **sp. nov.**

(*): *Vincx (1983)*, redescribed *D. schulzi* and mentioned that this species presents "false lateral alae" and 12 longitudinal rows of hair-like spines, following terminology of *Verschelde, Gourbault & Vincx (1998)*.

(**): the original description of *D. tenuispiculum* lacks data on longitudinal ridge number. Reported counts: 12–20 in *D. cephalata* sensu *Chitwood (1936)*, 16 by *Gerlach (1950)*, 24 by *Gerlach (1963)*, 12 by *Boucher (1975)*, 15 by *Platt & Warwick (1988)*, and 16–18 by *Fadeeva, Mordukhovich & Zograf (2016)*.

## DISCUSSION

*Gerlach (1950)* described *Desmodora schulzi* and, years later in his revision (*Gerlach (1963)*, synonymized *Desmodora schulzi* with *Heterodesmodora hirsuta Chitwood, 1936*, establishing the new combination *Desmodora hirsuta* (*Chitwood, 1936*) *Gerlach, 1950*. *Vincx (1983)* redescribed *Desmodora schulzi*, disagreeing with *Gerlach (1963)* proposed synonymy and indicating the characteristics that differentiate *Heterodesmodora hirsuta* from *Desmodora schulzi*. Later, *Verschelde, Gourbault & Vincx (1998)* transferred both species to the genus *Desmodorella*, considering them valid and distinct from each other. We agree with *Vincx (1983)* and *Verschelde, Gourbault & Vincx (1998)* in treating *Desmodorella hirsuta* and *Desmodorella schulzi* as distinct species, and therefore not synonymous. When comparing females of both species, it is possible to note that, with the exception of the total body length and the de Man ratio "c", other features and body proportions differ (see the comparison between these taxa in the discussion section in *Vincx (1983)*). Additionally, although both species share the number of longitudinal rows of spines (*D. hirsuta*: 10 rows; *D. schulzi*: 8–10 rows), this feature is not sufficient to synonymize the species. Similar to the aforementioned species, *D. sanguinea* also has 10 longitudinal rows of spines, and is easily distinguished from *D. schulzi* by comparing the characteristics present in males (spicules length, morphology of the gubernaculum and precloacal supplements). However, we disagree with *Vincx (1983)* and *Verschelde, Gourbault & Vincx (1998)* regarding the validity of *D. hirsuta*. Since this species was described based on a female (*Chitwood, 1936*), making it difficult to distinguish it from other *Desmodorella* species, we believe that there is no sustainable evidence to consider it as a valid species. Here, we formally suggest that *Desmodorella hirsuta* (*Chitwood, 1936*) *Verschelde, Gourbault & Vincx, 1998* be regarded as a *nomen dubium*.

To develop the dichotomous key, the main characteristics that, together, effectively helped distinguish the *Desmodorella* species were: morphology and number of turns of the amphidial fovea; spicules length (short or elongated) and morphology; the presence or absence of vacuoles in the head capsule, as well as in the rings along the body; the presence

or absence and morphology (elongated or short) of the subcephalic setae; the presence or absence of two pairs of lateral rows of more distinct spines, among the other rows of spines (referred to by *Verschelde, Gourbault & Vincx (1998)* as "false lateral alae"); number of longitudinal rows of somatic setae; number of longitudinal rows of ridges or spines; and morphology of the precloacal supplements. Although these are relevant characteristics for species identification/differentiation, the presence or absence of subcephalic setae and the number of longitudinal rows of ridges or spines should be analyzed with caution. Subcephalic setae can be lost during specimen preparation, and their presence, in some cases such as in *D. filispiculum*, is inferred from the visualization of the insertion point of the setae (*Lorenzen, 1976*). The number of longitudinal rows of ridges or short spines can often be difficult to determine, especially through optical microscopy, as mentioned by *Verschelde, Gourbault & Vincx (1998)* when describing *D. balteata*. Despite providing SEM analyses, *Verschelde, Gourbault & Vincx (1998)* did not mention the number of longitudinal rows of spines that occur in *D. balteata* and *D. spineacaudata*. When redescribing *D. tenuispiculum, Fadeeva, Mordukhovich & Zograf (2016)* reported that the visualization of rows was only possible through SEM analyses. Although it was possible to visualize the rows of ridges in some paratypes of *D. parabalteata* **sp. nov.**, the SEM analysis allowed us to demonstrate the configuration of these structures more clearly. However, it was not possible to precisely determine the number of rows that occur in this species, with variation in the number of rows along the body (a higher number of rows in the widest part compared to the median region where the body narrows), along with the occurrence of discontinuous rows. The literature on *D. tenuispiculum* records a large variation in the number of longitudinal rows of ridges present in this species. When describing *D. tenuispiculum*, *Allgén (1928)* did not indicate the number of rows present in the species. In subsequent redescriptions, the number of rows varied between 12 and 24 (*Chitwood, 1936*; *Gerlach, 1950*; *Gerlach, 1963*; *Boucher, 1975*; *Platt & Warwick, 1988*; *Fadeeva, Mordukhovich & Zograf, 2016*). Similarly, when redescribing *D. schulzi*, *Vincx (1983)* reported the presence of 12 longitudinal rows of hair-like spines along the body, while the original description (*Gerlach, 1950*) reported that there are 8–10 rows. These variations may be due to the difficulty in visualizing and determining the number of rows or may reflect an intraspecific variation regarding this feature. Therefore, it is extremely important that the characteristics found in *Desmodorella* species are analyzed together to determine and identify the species.

*Desmodorella cornuta* **sp. nov.** possesses a protuberant horn-shaped cuticular projection positioned dorsally in the pharyngeal region. This feature is unique among the *Desmodorella* species but can be observed in the Desmodoridae genus *Spinonema Larrazábal-Filho et al., 2019*. This genus encompasses species that possess a strongly cuticularized dorsal spine located in the pharyngeal region. However, *Spinonema* species have C-shaped anteriorly oriented lateral alae (without spines) and spicules that may possess a velum. *Desmodorella cornuta* **sp. nov.**, on the other hand, has two pairs of lateral rows with more distinct spines, among the other rows of spines and spicules lacking velum, a combination of characteristics typically found in representatives of the genus *Desmodorella*. The genus *Spinonema* was originally described from specimens found in sediment samples collected in the Potiguar Basin, northeastern coast of Brazil, the same type locality as *Desmodorella*

*cornuta* **sp. nov**. We believe that the occurrence of similar structures in different genera of Desmodoridae may reflect a process of adaptive convergence. The occurrence of a protuberant horn-shaped cuticular projection in the pharyngeal region was included in the diagnosis of the genus.

This study enhances current knowledge of *Desmodorella* biodiversity, introduces new diagnostic features, updates the list of valid species, and highlights key morphological traits that should be jointly considered for accurate species identification within the genus.

## ACKNOWLEDGEMENTS

The authors express their sincere gratitude to the project titled 'Evaluation of benthic and planktonic biota in the offshore portion of the Potiguar and Ceará basins' conducted by the Brazilian oil company Petrobras, particularly to its coordinator, Professor Dr. Paulo Jorge Parreira dos Santos from UFPE, for granting them the opportunity to examine the material associated with this project. We express our gratitude to the Departamento de Zoologia, UFPE for the scanning electron microscopy facilities at Laboratório Avancado de Microscopia e Imagem (LAMI-UFPE) of the Núcleo de Prospeccão e Gestão da Biodiversidade do Nordeste, Universidade Federal de Pernambuco. The Brazilian navy provided logistical support for the scientific cruise aboard the R/V Vital de Oliveira. We would like to thank Dra. Hianna Arely Milca Fagundes Silva, UFPE, for the technical support in obtaining the SEM photographs. The authors express their gratitude to the IAM-FIOCRUZ Core Facilities for providing access to Electron Microscopy Services.

### Funding

This study was financed by the Coordenacão de Aperfeiçoamento de Pessoal de Nível Superior Brasil (CAPES)_Finance Code 001. Alex Manoel was supported by a FACEPE graduate scholarship (IBPG-1516-2.00/21). The funders had no role in study design, data collection and analysis, decision to publish, or preparation of the manuscript.

### Grant Disclosures

The following grant information was disclosed by the authors:
Coordenacão de Aperfeiçoamento de Pessoal de Nível Superior Brasil (CAPES)_Finance Code 001.
FACEPE graduate scholarship: IBPG-1516-2.00/21.

### Competing Interests

The authors declare there are no competing interests.

### Author Contributions

- Alex Manoel conceived and designed the experiments, performed the experiments, analyzed the data, prepared figures and/or tables, authored or reviewed drafts of the article, and approved the final draft.

- Patricia F. Neres conceived and designed the experiments, performed the experiments, analyzed the data, prepared figures and/or tables, authored or reviewed drafts of the article, and approved the final draft.
- André M. Esteves conceived and designed the experiments, performed the experiments, analyzed the data, prepared figures and/or tables, authored or reviewed drafts of the article, and approved the final draft.

## Data Availability

The data is available in the tables.

## New Species Registration

The following information was supplied regarding the registration of a newly described species:

Publication LSID: urn:lsid:zoobank.org:pub:0EC65900-F3B5-4486-B067-5721DAC18C4D

Desmodorella cornuta LSID: urn:lsid:zoobank.org:act:057ED89B-11DD-4188-B98C-AF39ACDFBC47

Desmoderella parabalteata LSID: urn:lsid:zoobank.org:act:A9593248-BF08-431D-8F75-21E307B9AC8B.

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
