# Peer review of "Two new free-living marine species of Desmodorella (Nematoda: Desmodoridae) from the continental shelf of northeastern Brazil, with an emended generic diagnosis and a dichotomous key to the species"

_PeerJ, doi:10.7717/peerj.20094_

## Round 0.1 · original submission · Minor Revisions

Dear Dr. Manoel, I ask you to make some minor corrections before the article can be published.

Reviewer 1 ·

Basic reporting

These scientific studies have interdisciplinary interest and fall within the scope of the journal. The topic is relevant for specialists in the field of parasitology, zoology and ecology. Therefore, the issues considered by the authors of the article require attention.
In the "Introduction", the authors describe the topic of the study. However, the authors of the manuscript do not indicate the issues that still remain unclear in this area. The purpose of the study is missing. Brief information about the role of these nematodes in the ecosystem can also be added to this section.

Experimental design

Sources are correctly cited. The review is logically organized into related sections and subsections. However, the list of references contains more than 50% outdated literature (before 2000). Therefore, it is desirable to increase the number of sources by adding more modern literature.

Validity of the findings

The introduction of the manuscript lacks a research objective. However, there is also no “Conclusion,” in which the authors should highlight their contribution to this area of research, as well as highlight unresolved issues and future directions.

·

Basic reporting

I have no major concerns at this stage. The manuscript, in its present form, generally meets the journal’s standards for publication. It demonstrates scientific clarity, appropriate organization, and adherence to formatting guidelines. While the content is sufficiently comprehensive, I recommend minor revisions to improve overall clarity and consistency. These suggestions have been provided directly within the annotated manuscript and should be addressed before final acceptance.

Experimental design

There are no issues to raise regarding the experimental design. The overall structure of the research has been thoughtfully and appropriately constructed. The design is logical and systematically laid out, allowing the reader to follow the progression of the study with ease. The authors have clearly articulated the central research question, which is both relevant and scientifically meaningful. Moreover, the methods employed to investigate this question are rigorous and well suited to the study’s objectives. Each step of the methodology has been executed in a manner that ensures reliability and reproducibility, reflecting a high standard of scientific practice.

Validity of the findings

I have no additional remarks at this point. The findings presented in the manuscript appear to be reliable and are thoroughly supported by the empirical data collected during the study. The conclusions drawn are logically derived from the results and remain well within the boundaries of what the data substantiate. Furthermore, the authors have successfully maintained a clear connection between their initial research objectives and the final interpretations, avoiding unwarranted speculation or overextension of the results.

Additional comments

The submitted manuscript demonstrates a high standard of academic writing, with the content presented in a clear and organized manner. It contributes meaningful and scientifically relevant taxonomic insights, which add value to the existing body of knowledge. The accompanying figures and tables are thoughtfully constructed, visually clear, and effectively reinforce the content discussed in the main text. At this stage, I do not have any further remarks concerning the overall structure or organization of the manuscript. Nevertheless, during my review, I have provided several targeted comments and proposed edits directly within the annotated version of the document. I encourage the authors to thoroughly review these notes and incorporate the recommended changes as needed to further enhance the clarity and accuracy of the revised manuscript.

Reviewer 3 ·

Basic reporting

Structure of the MS is clear, renevant literature is appropriately referenced, line drawings and pictures are of a high quality.
English language of the MS is for quite well to my mind, but I am not a mative speaker.

Experimental design

Aims and scope, research question are well defined. Methods are described in detail.

Validity of the findings

Novelty is evident. All the necessary data are provided. Conclusions are well stated.

Additional comments

This is a very solid and conscientious study, and well written manuscript. I have only a few minor remarks.

REMARKS
Line 85. The authors state that the genus [Desmodorella] was identified using keys by Warwick et al. (1998). However, this volume refers to Monhysterida, not to Chromadorida & Desmodoroda. Is it a mistake?
Line 233. Description of Desmodorella cornuta. “Buccal cavity with … a small ventrosublateral tooth”. If possible, specify position of the tooth, right of left.
Line 234. “Nerve … not observed” Obviously, you mean nerve ring?
Line 266. (Jensen, 1985) is missing in References.
Line 431-432. Meanings 10-14 longitudinal rows … in thesis and 12-24 longitudinal rows … in antithesis are overlapped and hence cannot be used in the dichotomous key.
Line 498. D. schulz has to be replaced by D. schulzi.
Line 552. Boucher, G. Year of publication is missing.
Line 585. Inglis WG… should be transferred to the position below according to the alphabet order.
Line 637. …mit extrem lange … to be replaced with … mit extrem langen …
Line 670. Correct name of the journal should be in full: Abteilung für Systematik, Geographie und Biologie der Tiere

---

## Round 0.2 · Minor Revisions

Dear Dr. Manoel, I kindly ask you to carefully correct the comments made by the reviewers before the article is approved for publication.

Reviewer 1 ·

Basic reporting

Scientific research has interdisciplinary interest. The article is written in professional English. The text of the manuscript contains all the necessary references to literary sources. The structure of the article corresponds to an acceptable format of «standard sections». The figures and tables of the manuscript are structured, sufficiently described and have all the notations for convenient visual perception.

Experimental design

The study is consistent with the objectives and scope of the journal. The manuscript clearly states the relevance of the study and its purpose. The research methods are described in detail.

Validity of the findings

The authors took into account all comments and made corrections. The conclusions were clearly formulated.

·

Basic reporting

The recommendations from the first round of peer review have been very well addressed. At present, there are no major concerns. The manuscript, in its current form, generally meets the journal’s standards for publication. It demonstrates scientific rigor, a well-structured framework, and adherence to formatting requirements. I have suggested a few additional revisions, but they are minor. These recommendations are noted in the annotated manuscript and should be incorporated prior to final acceptance.

Experimental design

The experimental design of this study is well-conceived and appropriately structured, with a clear and logically organized progression. The authors have effectively presented the central research question, which is both highly relevant and scientifically meaningful. Moreover, the methods employed to address this question are robust and well aligned with the objectives of the study. Each methodological step has been carefully executed to ensure reliability and reproducibility, reflecting a high level of scientific rigor.

Validity of the findings

At this point, I have no further comments. The results presented in the manuscript appear to be reliable and are well supported by the empirical data collected during the research. The conclusions are logically derived from the findings and are appropriately framed within the limits of what the data can substantiate.

Additional comments

This manuscript demonstrates a high level of taxonomic significance, with its content presented clearly and in a well-organized manner. The study makes a meaningful and scientifically valid taxonomic contribution that adds value to the existing literature. The figures and tables in the manuscript are thoughtfully prepared, visually clear, and effectively illustrate the content discussed in the main text. At this stage, I have no further comments regarding the overall structure or organization of the manuscript. However, during the second round of review, I have indicated a few minor additional revisions as annotations in the manuscript. I recommend that the authors carefully review these suggestions and incorporate the recommended changes accordingly.

Reviewer 4 ·

Basic reporting

The authors present the description of two new species of free-living marine nematodes from continental shelf of Northeastern Brazil.
I am evaluating already a reviewed version of the manuscript but I had opportunity to read comments of previous reviews and see that they recommended few corrections. I agree with previous comments. The descriptions are very well done and the figures and tables have high quality.
I have only few suggestions, mostly some textual and type corrections.
For all these reasons I suggest to accept the manuscript for publication after minor corrections.

Experimental design

Experimental design is correct. Material and Methods is well described.

Validity of the findings

The authors describe two new species. The descriptions are well presented. I am convinced that they are indeed new species.

Additional comments

Line 18: delete “other”
Line 31: The correct way to cite Nemys is “Nemys eds. (2025)”. This is clearly explained in Nemys web site. Please correct this in all citations through the manuscript (i.e. line 155).
Line 41: Please use “Gerlach (1963)”
Line 128: Decraemer & Smol (2006) did not propose a taxonomic classification. The correct is refer to De Ley & Blaxter (2004) classification.
Line 137-150: Please add also female reproductive system characteristics to diagnosis (i.e. number of ovaries, reflexed or not…).
Lines 172 and 183: authors of the new combinations are missing after parentheses. Please add!!! This is recommended by the Code!
Lines 198-209 and lines 280-291: Please add environmental characteristics to type locality description (i.e. sediment granulometry).
Line 214: something is missing in the end of sentence… maybe the correct is “widest at level of testis”?
Line 229: the outer labial sensilla is either papillous or setiform… I believe here is only “papillous” as in the first circle…
Line 231: please write “posterior part of head capsule”
Line 240: correct to “elongated”
Line 252: correct to “elongated”
Lines 252 and 333: females have anal diameter (not cloacal!)
Line 300: always use “spicules” (in plural!) when the spicules have the same morphology. This should be corrected in many places of the manuscript (including tables!). Please revise carefully! (i.e. lines 366, 367, 385, 387, 457; Tables 2, 3 and 4…)
Line 318: please correct to “ Reproductive system monarchic with anterior outstreched testis on left…”
Lines 376-379: the correct is “spicules longer” (instead of greater). Then “long” can be deleted in the end of sentence (i.e. Spicules longer than 100 micrometers).
Line 385: use “shorter” (instead of less).
Line 435: author of the new combination is missing after parentheses. Please add!!! This is recommended by the Code!
Line 476: use “higher” (instead of greater)
Line 492: write all authors in extend here. We do not use “et al” for species authors. This is explained by the Code.

Figure Legends:
Figure 5. There is no “hspc projection” in 5B! Please delete!
Figure 7. Please correct to “cephalic setae”.

---

## Round 0.3 · Minor Revisions

I ask you to make a few minor corrections before the article is accepted for publication.

·

Basic reporting

no comment

Experimental design

no comment

Validity of the findings

no comment

Additional comments

Thank you for carefully following the reviewer’s instructions regarding the revisions. Finally, I have marked a few minor corrections directly on the manuscript. Please review and make the necessary changes. There are no further revisions.

---

## Round 0.4 · accepted · Accept

Dear Dr. Manoel, I congratulate you on the acceptance of this article for publication.